# The Use of Foetal Doppler Ultrasound to Determine the Neonatal Heart Rate Immediately after Birth: A Systematic Review

**DOI:** 10.3390/children9050717

**Published:** 2022-05-13

**Authors:** David Hutchon

**Affiliations:** Industry and Innovation Research Institute, Sheffield Hallam University, Sheffield S1 1WB, UK; david.hutchon@student.shu.ac.uk

**Keywords:** neonatal, heart rate, resuscitation, Doppler, precordial

## Abstract

Determining the neonatal heart rate immediately after birth is unsatisfactory. Auscultation is inaccurate and provides no documented results. The use of foetal Doppler ultrasound has been recognised as a possible method of determining the neonatal heart rate after birth over the last nine years. This review includes all published studies of this approach, looking at accuracy, speed of results, and practical application of the approach. Precordial Doppler ultrasound has been shown to be as accurate as ECG and more accurate than oximetry for the neonatal heart rate, and provides quicker results than either ECG or oximetry. There is the potential for a much improved determination and documentation of the neonatal heart rate using this approach.

## 1. Introduction

During labour, the main parameter for determining the health of the foetus is heart rate (HR). After birth, when the newborn baby can be observed, heart rate is still a major measure of health—especially in a compromised neonate who is not breathing, crying, or moving. The neonatal branch of the International Liaison Committee on Resuscitation recommends that the heart rate of the newborn baby be determined within the first minute after birth. Heart rate determines the level of care required and any interventions. Auscultation is recommended for routine measurement of the heart rate, which needs to be determined quickly and accurately, since there are specific thresholds for intervention defined by the neonatal heart rate [1].

Determining the heart rate by auscultation requires the number of heartbeats heard to be counted over a known interval of time. The interval needs to be short—6 s or 10 s is recommended—so as to quickly obtain the estimated rate in beats per minute. In healthy neonates, auscultation provides a satisfactory estimate of the heart rate as determined by an ECG, but in a compromised neonate, in a noisy birth room and with weak heart sounds, this may not be the case [2]. Furthermore, the heart rate is undocumented in real time, and there is no opportunity to review the result.

For these reasons, other approaches to determine the heart rate have been sought. Oximetry technology is widespread, reliable, and low-cost. It is the standard for determining the health of neonates. ECG technology is also widespread, and increasingly reliable and low-cost. Both of these technologies take time to apply, and may be difficult after the neonate is placed in a polythene bag—routine care in preterm neonates to reduce the risk of hypothermia. Both of these technologies can readily record and document the heart rate for later review.

A more satisfactory method for routine use to determine the neonatal heart rate at birth is required. Doppler ultrasound is the routine method for monitoring the foetal heart rate. Ultrasound readily passes through the maternal tissues, and the frequency of the reflected sound is altered by the movements of the heart. Heart valve movements are rapid, and result in the major Doppler effects. Foetal heart sounds are familiar to all midwives and obstetricians. The strength of the sound is correlated with the speed of the tissue movement. The heart rate can be determined through an electronic algorithm, or by the clinical assessment of counting over a measured time. The safety of Doppler ultrasound is fully established, with its use in millions of pregnancies over the last 50 years.

Recently, the possibility of this well-established technology used for the foetal heart rate has been explored for use in determining the neonatal heart rate, and the use of Doppler ultrasound forms the basis of this systematic review.

## 2. Materials and Methods

A literature search using the Medline database with the search terms “neonatal”, “doppler”, and “heart rate” in the title showed 8 publications; 2 were excluded as not relevant, and a further paper was a review with no new data. A second Medline search using the terms “neonatal”, “heart rate”, and “resuscitation” in the title identified 24 results. From this, one further paper was identified that described the use of Doppler ultrasound for determination of the neonatal heart rate under “novel techniques”. Additional published articles were obtained by manually searching the references in the above publications (Table 1).

## 3. Results

Eleven publications were identified as possibly relevant (Table 2). One study by Dyson et al. describing the use of vascular Doppler of the aorta to determine the neonatal heart rate [8], as well as another by Lemke et al. [9] measuring blood flow of the umbilical stump, were excluded (Table 3). A review paper containing no new original data was excluded [2]. Of the remaining studies, five were in human subjects and two in piglets. One excluded study referred to the use of precordial Doppler at birth in the text, but provided no data [10].

Agrawal et al. and Kayama et al. included a range of both healthy and asphyxiated neonates at term or preterm [5,7]. Zanardo et al. [3], Shimabukuro et al. [4], and Goenka et al [6]. focused on healthy neonates. Of the two animal studies, one was in healthy newborn piglets [11], while the other was in asphyxiated piglets [12]. These studies were not included in the data analysis, but are included in the Discussion, for the reasons explained.

The foetal Doppler machines used in the studies were either 2 or 3 MHz transducers. All were handheld transducers placed over the neonatal precordium with a sound output and a digital heart rate display. The gold standard for the heart rate in the studies was the electrocardiogram (ECG), and in addition, pulse oximetry was measured in one study. All studies showed a satisfactory correlation of the displayed heart rate between Doppler and the gold standard.

The time taken after application of the Doppler probe for a digital heart rate display was quite variable between the studies, but was less than or equivalent to the time taken for the ECG to display a heart rate.

In one study, the authors found that the Doppler ultrasound did not function well with crying and moving babies [7].

## 4. Discussion

The current clinical approach for determining and documenting the heart rate of a baby immediately after birth is not satisfactory. It is inaccurate, and it is usually more than one minute after birth before the heart rate is obtained [2]. To guide further intervention in compromised neonates, the attendant needs to know whether predefined HR targets have been reached. The effectiveness of positive pressure ventilation (PPV) is determined by an increase in the neonatal HR.

The findings from the piglet studies are included in this review because they demonstrate that the Doppler approach can still detect the heart rate even in very preterm neonates of 500 g—the same weight as a typical newborn piglet. Morina et al. also showed that Doppler ultrasound worked satisfactorily in severely asphyxiated piglets [12]. Hutchon showed the heart rate could be determined and documented within seconds after birth even in these tiny animals [11].

### 4.1. Accuracy and Time Taken to Obtain the Heart Rate

Auscultation is the recommended mode to determine the initial heart rate in newborns by ILCOR [1]. It takes no significant amount of time to place the stethoscope on the neonate’s chest. However, thereafter, methods for getting an accurate heart rate are unsatisfactory. Auscultation is intermittent, and heard by only one clinician, who has to count the number of heartbeats over a fixed time period. Inevitably, this method has limited accuracy, and is undocumented for later review and audit. Background noise in the delivery room may further interfere with the audible heart sounds—especially with the low-level heart sounds of a compromised or preterm neonate.

The application of an ECG (2–3 electrodes) or pulse oximeter takes considerably longer than the application of a stethoscope. The application of the Doppler, however, should be equivalent to applying the stethoscope.

The long latency period of 1 to 2 min before a reliable signal display is obtained from the ECG and PO is a serious shortcoming, as this long interval may interfere with optimal care of the neonate during the first minutes after birth. Some of this time is taken to apply the oximeter sensor or ECG electrodes—longer if three rather than two electrodes are necessary (depending on the equipment being used).

The ECG is considered the gold standard for the accurate measurement of the heart rate (with the previous qualification). Algorithms within the equipment determine the interval between two QRS complexes to measure the heart rate. There are similar algorithms within the Doppler and oximetry electronics. The time taken to display a result, and the result’s precision, are therefore very dependent upon the make, design, and age of the Doppler, ECG, or oximetry electronics, but all equipment used in these studies in the last 10 years can be considered satisfactory.

### 4.2. Safety

All four methods of neonatal heart rate have an established safety record. Intrapartum monitoring (CTG) with Doppler ultrasound has been used in millions of births throughout the world over the last 30 or more years. The ultrasound gel used to facilitate the transmission of ultrasound from the transducer to the neonatal chest is routinely used in colour Doppler examination of the neonatal heart, and there have been no adverse effects reported. However there have been concerns about the ECG electrodes—especially when applied for longer than a few minutes—causing pain or injury to the newborn’s skin [5].

Hypothermia is a concern at birth with the exposed wet newborn skin. Babies are therefore routinely dried and wrapped in towels, although an exposed newborn baby can be safely cared for under an infrared lamp present on the neonatal resuscitation trolley. After application of the ECG electrodes and pulse oximeter, the baby can be covered. Application of the precordial Doppler, on the other hand, requires exposure of the chest. However, exposure of the chest may also be necessary during resuscitation to check for the entry of air to the lungs, or to observe chest movements. For a preterm neonate placed in a polythene wrap to reduce the risk of hypothermia, the Doppler ultrasound will still function through the polythene.

Auscultation requires the head of the stethoscope to be placed on the neonate’s chest with sufficient pressure to provide good contact with the neonatal skin, but avoiding any excessive pressure that could interfere with chest’s expansion during breathing. In practice, avoiding excessive pressure may be quite difficult. The same limitation applies to the precordial handheld Doppler.

### 4.3. Resuscitation

In a baby who is breathing, crying, or moving, determining the heart rate immediately after birth is not a priority, and Kayama et al. showed that the application of Doppler ultrasound is sometimes difficult in these babies [7]. However, auscultation is likely to be just as limited, and the precise heart rate is of little importance in these healthy babies. It is in hypotonic, apnoeic neonates—likely to require assistance to transition from placental to pulmonary respiration—that there is some urgency to know the heart rate, and whether it is increasing or decreasing [13]. Knowledge of the accurate heart rate of a hypotonic, apnoeic neonate by the whole resuscitation team is important. Ideally, this needs to be a continuous result from soon after birth, and fully documented for subsequent case review. If the Doppler heart sounds are heard by the whole team, they can make estimates of the rate and whether or not it is increasing. Some estimates of the strength of the cardiac contraction may be possible from the Doppler sound.

Pulse oximetry and neonatal ECG are both established methods for determining the heart rate in newborns. However, both have limitations. Any one method may not be ideal, and ECG, oximetry, and Doppler are likely to complement one another in severely compromised neonates. Oximetry requires a significant capillary blood flow, and does not function during the first minute or so after birth even in a healthy neonate. Because it requires a good capillary blood flow, it may not function well in compromised neonates until after resuscitation has been successful. Typically, it takes over a minute after birth for the oximeter to register a heart rate, even in healthy-term neonates. This delay is likely to be significantly longer in compromised neonates, especially when the response to resuscitation is not immediate. A visible oximetry tracing is sometimes available for the resuscitation team, but this requires the attendants to look away from the neonate. ECG can also provide both a visible tracing and digital heart rate, but this requires the attendant to look away from the neonate.

ECG depends on the electrical activity of the heart and, at least in moribund newborns, the possibility of pulseless electrical activity (PEA) needs to be considered. PEA has in fact been identified through the use of colour Doppler flow [14], which is essentially a graphic form of Doppler ultrasound. It could therefore be argued that Doppler ultrasound is actually the gold standard.

### 4.4. Sterility

Many compromised neonates are delivered by caesarean section, and there is increasing recognition of the importance of resuscitation—when required—with an intact umbilical cord. In healthy neonates, a period of up to three minutes before the cord is clamped is increasingly adopted, and the heart rate needs to be determined during this time [15]. This may require the neonatal resuscitation team to be within or very close to the sterile operating field. It is possible to use a sterile stethoscope to determine the neonatal heart rate in this scenario; however, the application of ECG electrodes or an oximeter presents a significant challenge. Pulse oximetry is in any case unsatisfactory in determining the heart rate during the first minute. However, Doppler ultrasound functions perfectly through polythene as a sterile cover, allowing the Doppler sound to be heard and the displayed heart rate to be seen by the team, and without any special adaption of the current equipment.

### 4.5. Future Development

A major advantage of ECG and oximetry is that they are hands-free after initial application. The possibility of a hands-free precordial Doppler system has been explored recently. A lightweight transducer that is connected to the Doppler machine by a soft, flexible wire makes the system hands-free. It will sit on the neonate’s chest as a result of the surface tension of the ultrasound gel alone, remaining in place even with some movement of the neonate [16]. If auscultation for the entry of air to the lungs or observation of the chest is required, the transducer can simply be lifted off, and later reapplied with a little more ultrasound gel.

## 5. Conclusions

The use of foetal Doppler to determine the neonatal heart rate at birth has recently been explored. Data are limited, but what evidence there is shows that it has the potential for a more accurate and quicker acquisition of the heart rate immediately after birth. Modifications could make the approach more user-friendly, and it is an established low-cost technology.

## Figures and Tables

**Table 1 children-09-00717-t001:** Human studies included in the review.

Study	Study Design	Comparisons
Zanardo and Parotto (2019) [3]	Observational study of 102 newborns (43 preterm) and 21 requiring resuscitation	ECG vs. 3 MHz Doppler
Shimabukuro et al. (2017) [4]	Prospective cross-sectional study of 33 term neonates at elective caesarean section	ECG vs. 3 MHz Doppler
Agrawal et al. (2021) [5]	Prospective multicentre study, 131 healthy neonates > 34 weeks	ECG vs. 2 MHz Doppler
Goenka et al. (2013) [6]	Prospective study of 92 stable newborns > 37 weeks, 1–8 min after birth	ECG and pulse oximetry vs. 3 MHz Doppler
Kayama et al. (2020) [7]	Prospective study of 102 newborns up to 72 h after birth, 21 during resuscitation, from 23 weeks to term gestation	ECG vs. 3 MHz Doppler

**Table 2 children-09-00717-t002:** Human studies included in the review.

Study	Heart Rate Accuracy	Interval from Application to Display
Zanardo and Parotto (2019) [3]	Good correlation between Doppler and pulse oximetry	Mean interval of 3 s for Doppler vs. 5.2 s for ECG
Shimabukuro et al. (2017) [4]	Good correlation between Doppler and ECG	5.6 s for Doppler vs. 5.2 s for ECG
Agrawal et al. (2021) [5]	Mean of 152 for Doppler vs. 161 for ECG	Significantly quicker for Doppler
Goenka et al. (2013) [6]	Good correlation between Doppler and ECG	Equivalent time to ECG and quicker than oximetry
Kayama et al. (2020) [7]	Good correlation between Doppler and ECG. A crying, moving baby may make Doppler measurements difficult	Mean of 5 s for Doppler vs. 10 s for ECG

**Table 3 children-09-00717-t003:** Excluded studies.

Dyson J et al. (2017) [8]
Lemke RP et al. (2011) [9]
Kevat AC et al. (2017) [2]
Katheria AC et al. (2017) [10]

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
