# Peer review of "The Use of Foetal Doppler Ultrasound to Determine the Neonatal Heart Rate Immediately after Birth: A Systematic Review"

_children, 2022, doi:10.3390/children9050717_

Round 1

Reviewer 1 Report

  1. The introduction could be shortened and the justifications for this review (pulse oximetry being slower and ECG being cumbersome) do not need to be repeated in the discussion
  2. I did not clearly understand how many total studies were actually reviewed - just the 5 studies in Table 1?
  3. I don't see why there is a Table for studies that are excluded
  4. How feasible is it to have a Doppler for HR detection during neonatal resuscitation? From the studies cited, I did not see that and that would be something readers would be interested in knowing. 

Author Response

  1. The introduction could be shortened and the justifications for this review (pulse oximetry being slower and ECG being cumbersome) do not need to be repeated in the discussion.       The introduction has been altered. "Both these technologies take more time to apply than a stethoscope for precordial auscultation . . ."  It is important that the need for an improved method for obtaining the neonatal heart rate at birth is fully explained and the limitations of current technology described. 
  2. I did not clearly understand how many total studies were actually reviewed - just the 5 studies in Table 1?      There are only five studies of the emerging use of precordial doppler in the neonate in the last nine years and none before this.
  3. I don't see why there is a Table for studies that are excluded    It is standard practice to list any potential studies which have been excluded from the initial search.
  4. How feasible is it to have a Doppler for HR detection during neonatal resuscitation? From the studies cited, I did not see that and that would be something readers would be interested in knowing.   Agrawal et al and Kayama et al included a range of both healthy and asphyxiated neonates at term or preterm [5,7].  There was in addition asphyxiated piglets which were referred to in the discussion.  "Morina et al also showed that doppler ultrasound worked satisfactorily in severely asphyxiated piglets [12]."  The theoretical advantage of doppler during resuscitation at caesarean section with an intact cord is also included in the discussion.

Reviewer 2 Report

This is a very exciting and well-written systematic review about another way to accurately determine postnatal heart rate, e.g. in neonates with impaired postnatal adaptation or resuscitation needs.

Especially since correct and rapid auscultation of the heart rate is often difficult in the delivery room (background noise), the use of fetal Doppler ultrasound seems an obvious alternative. In particular, children with pulseless electrical activity can thus also be detected. The use in neonatal care at the umbilical cord, e.g. in the immediate surroundings of the sterile surgical field, could also be another possible application.

The discussion of the advantages and disadvantages or the various aspects of Doppler examination compared to ECG and oxygen saturation is well presented and detailed.

The following aspects might be added:

Results:

For the human studies (or animal studies) listed in Table 1 that were included in the review, I would have been interested in more details in the results section, if necessary, in tabular form to better assess these studies (i.e., study design, term infants/preterm infants, number of infants studied per study, exactly which techniques were studied or compared in each study). Information on whether the infants were healthy or more severely asphyxiated infants could also be presented in such a table.

Author Response

For the human studies (or animal studies) listed in Table 1 that were included in the review, I would have been interested in more details in the results section, if necessary, in tabular form to better assess these studies (i.e., study design, term infants/preterm infants, number of infants studied per study, exactly which techniques were studied or compared in each study). Information on whether the infants were healthy or more severely asphyxiated infants could also be presented in such a table.  An additional table has been added.